# Enforcing Object Permanence using Hierarchical, Object-Centric Generative Models

**Toon Van de Maele**
IDLab, Ghent University - imec
`toon.vandemaele@ugent.be`

**Stefano Ferraro**
IDLab, Ghent University - imec
`stefano.ferraro@ugent.be`

**Tim Verbelen**
IDLab, Ghent University - imec
`tim.verbelen@ugent.be`

**Bart Dhoedt**
IDLab, Ghent University - imec
`bart.dhoedt@ugent.be`

## Abstract

Object permanence is an important milestone in infant development, when the infant understands that an object continues to exist even when it no longer can be seen. However, current machine learning methods devised to build a world model to predict the future still fail at this task when having to deal with longer time sequences and severe occlusions. In this paper, we compare current machine learning with infant learning, and propose an object-centric approach on learning predictive models. This grounds object representations to an inferred location, effectively resolving the object permanence problem. We demonstrate performance on a novel object-permanence task in a simulated 3D environment.

## 1 Introduction

An important milestone in infant development is to understand object permanence (Piaget, 1954). This is the ability to know that objects continue to exist, even when they are hidden from sight. From a neuroscience perspective, learning is often cast as the brain minimizing prediction error, also coined free energy (Friston et al., 2016). This appeals to machine learning approaches, in which parameterized models such as variational autoencoders (VAE) can be trained for predicting future sensory observations, effectively building a model of the world (Çatal et al., 2020; Hafner et al., 2020). Despite impressive results in learning representations in various domains, we found that current implementations are still unable to capture longer term dependencies, and hence fail at simple object permanence tasks.

To address this problem, we propose two novelties for learning such world models, inspired by human learning. First, we devise a hierarchical structure of the generative model, in which on the one hand, inferences are made about the kind of objects that are present, and on the other hand a global scene representation is constructed on where these objects are located in 3D space. This is consistent with recent accounts of the human vision system (Parr et al., 2021) which considers perception as inferring a scene as a factorization of separate (parts of) objects, their identity, scale and pose. Distinguishing scale and pose also matches the two stream hypothesis, which states that visual information is processed by a dorsal ("where") stream, and a ventral ("what") stream (Mishkin et al., 1983). Also, recent findings in recordings of rhesus monkey brains provide evidence that indeed 3D shape is encoded in the inferior temporal cortex (Janssen et al., 2000).

Second, we adapt the data regime with which these object-centric models are trained. In machine learning, representations are typically learned from large datasets of images (Grill et al., 2020) or

4th Workshop on Shared Visual Representations in Human and Machine Visual Intelligence (SVRHM) at the Neural Information Processing Systems (NeurIPS) conference 2022. New Orleans.

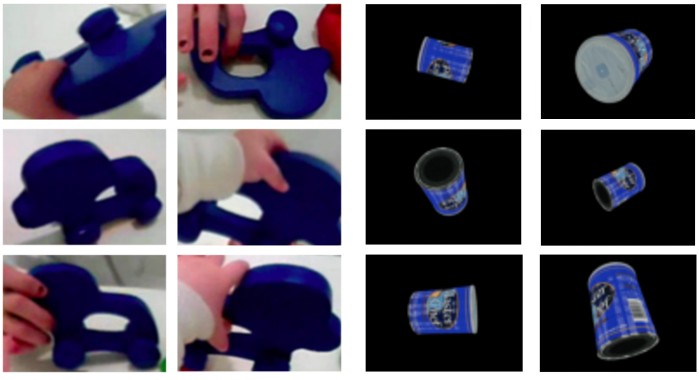

|                          |                            |
|:------------------------:|:--------------------------:|
| (a) toddler object views | (b) simulated object views |

Figure 1: (a) First person views of a 15 month old toddler are typically centered around a single object (adapted with permission from (Slone et al., 2019)). (b) Object-centric views sampled from our dataset for training the generative model.

videos (Qing-Yuan Jiang & Li, 2019) of a wide variety of objects and scenes. This is in stark contrast to toddlers, who learn from very object-centric views, by actively engaging with particular objects, as shown on Figure 1a (James et al., 2014; Slone et al., 2019). In similar vein, we therefore also train our object-centric models through novel view predictions of a single object instance in different poses (Van de Maele et al., 2022), as illustrated in Figure 1b. This way, our agent can infer distinct objects and their location in space, and maintain this knowledge to predict other viewpoints, even after complete occlusions. In the remainder of this paper, we summarize our approach in Section 2, show some preliminary results of this system in Section 3, and discuss some related work and future work in Section 4.

## 2 Object-Centric Generative Models

Our method is based on object-centric generative models that learn latent representations of object pose and identity (Van de Maele et al., 2022). Such models consist of an encoder, which maps pixels to an object identity and pose latent, a decoder which decodes identity and pose to a novel pixel observation, and a transition model which learns how the pose latent changes after moving the camera with an action. In addition to the encoder described in Van de Maele et al. (2022), we first predict the mask using $m_\phi$ and mask the observation before feeding it to the encoder. Learning to filter out anything that is not part of the object makes it more resilient against occlusions and variations in the background. Specifically, we model the following components using deep neural networks:

Mask: $\quad \mathbf{x}_{t,m} = m_\phi(\mathbf{x}_t),$      Transition: $\quad p_\phi(\mathbf{p}_t|\mathbf{p}_{t-1}, \mathbf{a}_{t-1}),$

Encoder: $\quad q_\phi(\mathbf{p}_t, i|\mathbf{x}_{m,t}) = q_\phi(\mathbf{p}_t|\mathbf{x}_{t,m})q_\phi(i|\mathbf{x}_{t,m}),$      Decoder: $\quad p_\phi(\mathbf{x}_t|\mathbf{p}_t, i),$

where $\mathbf{x}_t$ represents a pixel-wise observation, and $\mathbf{x}_{t,m}$ after masking. $\mathbf{p}_t$ represents the object-centric pose latent at time $t$, capturing the orientation and scale of the object in this timestep. $i$ is a latent variable, encoding the identity of the observed object, implemented as a Bernoulli variable. Finally, $\mathbf{a}_t$ is the action the agent performed at timestep $t$, represented as a relative translation and orientation, w.r.t. the current observation. The $\phi$ subscript represent the learnable parameters.

These models are trained end to end using a prerecorded dataset of an agent "playing around" with an object, i.e. capturing different views of the object, and predicting the next view. The dataset consists of renders from simulated meshes from (fixed-size) objects in the YCB dataset (Calli et al., 2015), which are augmented with random background colors and random occlusions of different negative objects. We optimize each model through the minimization of the negative evidence lower bound (ELBO). The reader is referred to the appendix for more details on optimization.

Next, we employ these models on pixel-based observations of a full scene containing multiple objects in separate distinct locations. We infer the different visible object identities and their corresponding poses in the scene. Using the predicted mask, for each object, we can infer the pixel coordinates

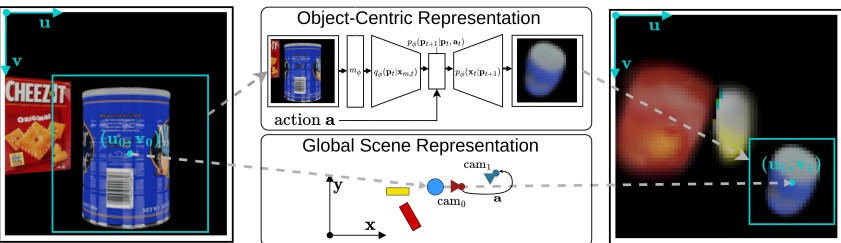

Figure 2: Visualization of our method for inferring a scene representation using object-centric models. Individual objects are detected using the object-centric models. Using the estimated pixel coordinates of the object within the image and a pinhole camera model, the 3D location of each object can be estimated. A novel viewpoint can now be rendered by estimating the pose and location of each object in the image, and compositing the predictions of the different object-centric models.

(uv-space) of the center for each object. Combining the pixel-coordinates with a pinhole model of the camera, we can now compute a belief over the object position using a 3D Multivariate Gaussian over the casted ray from the camera pose to the object center. Over multiple timesteps, information of each frame is aggregated using a Bayesian belief update rule. We call this collection of identities and their inferred positions the "Global Scene Representation" as illustrated in Fig. 2. For each of the detected objects, we also have an "Object-Centric Representation" - representing the orientation and scale of the object - acquired through encoding an object-centric crop of the object with their respective object-specific model.

To imagine a novel view, the camera pose is first acquired by applying the action to the current camera pose. Then, for each of the objects, an inverted projection is applied to acquire the pixel coordinate where the center of the object would be observed. For each of the objects, the action is applied to their transition model, and using the likelihood, a novel observation is generated. These are then composed around the predicted object center on the novel image canvas. The z-index of each object is acquired through the estimated distance of the object to the camera.

## 3   Object Permanence

To evaluate the ability of object permanence, we created a 3D simulation environment in which three objects from the YCB dataset (Calli et al., 2015) are rendered at distinct locations. The model is first given the first 16 images, in which an object becomes occluded due to the camera moving (Fig. 3.a). The goal is then to predict, given the consecutive actions, the next 16 images in which the camera moves further and the initial object becomes back in view (Fig. 3.b).

We compare our approach with a sequential VAE model (SSM) (Çatal et al., 2020), as well as a recurrent state space model (RSSM) as proposed by Hafner et al. (2020). In contrast to our approach, these models are trained on sequences of observations, predicting the next observation given a latent state inferred from previous observations and actions. In order to correctly predict future states, these models use a recurrent memory unit, i.e. an LSTM (Hochreiter & Schmidhuber, 1997) or GRU (Cho et al., 2014) respectively. To compare their performance, we train those on a dataset of similar sequences of actions and observations as used during evaluation.

Fig. 3 shows our results. While the final object position and pose are not perfect, our model is able to estimate the object presence over time (c). In the SSM and RSSM model, this is not the case and the model does not remember any of the objects, and just imagines a novel view.

Quantitatively, we evaluate our approach for permanence using an adapted mean squared error (MSE), computed only over the pixels of the earlier occluded object on a test set consisting of 1000 randomly generated sequences, which we can compute as we acquire the object mask directly from our environment. Since the distance of the agent to the object has a large impact on the error's order of magnitude, we scale this metric with the inverse amount of visible object pixels. We show these results in Table 1. In addition, we report a detection rate, which determines whether or not the agent has captured the correct object in the final frame. For this we count the amount of pixels within the object mask, where the error is lower than an empirically determined threshold of $0.20$, after which

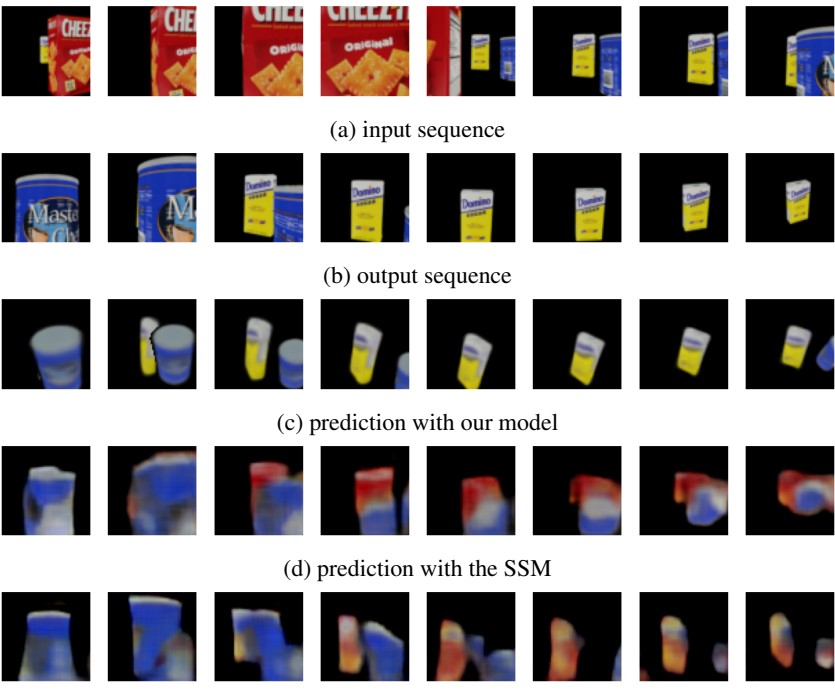

(a) input sequence

(b) output sequence

(c) prediction with our model

(d) prediction with the SSM

(e) prediction with the RSSM model

Figure 3: Sequence of observations where an object is occluded, together with predictions by the different models.

we count the objects where more than 25% of the object is correctly reconstructed. We observe that our approach has the lowest adapted MSE and the highest object detection rate of the compared models, which further supports our claims from the qualitative results.

## 4 Discussion

This work is related to other object-centric approaches for scene understanding. Most of these methods stem from the seminal work of Attend Infer Repeat (Eslami et al., 2016), where an observation is decomposed into object-level observations by having a recurrent neural network predicting the parameters of a spatial transformer network (Jaderberg et al., 2015). MONet (Burgess et al., 2019) on the other hand is trained entirely end-to-end to first recurrently predict a mask for each object present in the scene, and then use a variational autoencoder to encode each masked observation. This is very similar to our approach, where we jointly learn a mask and pose representation for each object, and then infer this separately for each object in the scene. However, we build a hierarchy on top that creates a "map" of the scene, representing the positions of the objects. In IODINE (Greff et al., 2020), a joint decomposition and representation model is learned with fixed slot-allocation. To scale this up to more objects, the encoders are adapted for predicting proposals in parallel instead of recurrently (Crawford & Pineau, 2019; Jiang et al., 2020).

Table 1: Quantitative comparison of the state space model (SSM), the recurrent state space model (RSSM) and our ensemble created from CCNs. We compare on the adapted MSE, a scaled MSE computed over the relevant pixels for the occluded object, and the percentage of scenes in which the agent is still able to reconstruct the correct object at the end of the sequence. This was run over 1000 scenes.

|                  | adapted MSE         | detected (%) |
|------------------|---------------------|--------------|
| SSM              | $0.4029 \pm 0.2653$ | 32.9         |
| RSSM             | $0.4466 \pm 0.2841$ | 37.3         |
| Scene CCN (ours) | $0.3535 \pm 0.2645$ | 60.0         |

When dealing with sequences of observations over time, most models learn to predict the dynamics of objects separately (Kosiorek et al., 2018; Jiang et al., 2020). (Kosiorek et al., 2018) implement this by adding a propagation module which predicts the movement of objects from previous timesteps. In addition, they also introduce a discovery module for detecting new objects when they appear. van Bergen & Lanillos (2022) investigate how object-centric models can be used within the active inference framework and extend the IODINE (Greff et al., 2020) model in a 2D d-sprites environment where forces can be applied on the sprites through action. In our work, we model action as 6-DOF movement of the camera in a 3D world, which not only adds complexity but also makes it much more relevant for real-world use cases such as robotics.

The most closely related work is ROOTS (Chen et al., 2021), where the authors consider multiple viewpoints from the same scene, crop out the objects of each viewpoint, and then group them together to encode them with a generative query network instance for each object (Eslami et al., 2018). In doing so, the objects can be rendered individually and aggregated in a full observation.

Developing commonsense is still an outstanding challenge in artificial intelligence (Davis & Marcus, 2015). Object permanence is a fundamental commonsense skill that emerges early during infant development (Piaget, 1954). Although recent advances in learning world models from sequences of observations and actions (Hafner et al., 2020), these still struggle on learning long-term dependencies. In order to improve long-term sequence predictions, one strategy is to add a hierarchy of latent sequences, where each level predicts at different time intervals (Saxena et al., 2021). This however is not trivial for action-conditioned scenarios, as there isn't necessarily an "action" construct at these higher, temporal levels (Zakharov et al., 2022).

A novel way of evaluating intuitive physics such as object permanence, is by evaluating whether or not an AI system renders an physically implausible trial as plausible, or vice versa (Moore et al., 2022). In similar vein, S. Punla et al. (2022) assess the Violation of Expectation (VoE) by measuring the surprise of the model. In addition, S. Punla et al. (2022) propose the PLATO model, which they demonstrate can learn a diverse set of physical concepts, among which is object permanence. Similar to our approach, PLATO also uses an object-centric representation, with a learned object code for each object. In contrast to our approach, they learn the object codes in an end-to-end fashion from sequential data, but using more simple primitives and a fixed camera viewpoint. Still, this illustrates that an hierarchical, object-centric scene representation might be key to develop spatial commonsense understanding.

Our approach uses a crop of the object to compute the pose, after which the action with respect to the center of the observation is applied. There will automatically be an offset as this action does not entirely match the movement with respect to the object. In cases the agent requires a precise estimate over the object pose, it is thus crucial to first move to a viewpoint where the object is central. While our approach does have a higher detection rate than the compared baselines, it uses an inductive bias that the scene is static and that the objects do not change over time. A potential extension would be to have a belief over object presence, which can be updated as well when evidence shows that an object has been removed from the expected position. In future work we plan to address these limitations and will work on methods where the higher level of the hierarchy is also learned end-to-end.

### Acknowledgments

This research received funding from the Flemish Government under the "Onderzoeksprogramma Artificiële Intelligentie (AI) Vlaanderen" programme.

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

# A    Appendix

## A.1    Optimization of the Object-Centric Models

These models are optimized using the minimization of the negative ELBO over a trajectory of the agent. In our implementation, we only consider sequences of one transition, and thus two timesteps ($T = 1$), as the environment is static and the agent can teleport/move to any viewpoint of the

object. For each timestep, a reconstruction error (MSE) is applied to both the reconstruction of the observation using the encoder model $p_\phi$ and the predicted mask $m_\phi(\mathbf{x}_t)$ directly, as well as a binary cross entropy over the predicted identity, which is equivalent to a KL-divergence between the approximate posterior and the marginal probability over identity $i$. The approximate posterior is conditioned on the masked input $\mathbf{x}_{t,m} = m_\phi(\mathbf{x}_t) \odot \mathbf{x}_t$. Starting from $t = 1$, the model can predict future observations and transition the latent pose variable. A reconstruction loss is also computed for the likelihood of each transitioned latent. Finally, a KL-divergence term draws the belief over the transitioned latent towards the belief over pose acquired through the approximate posterior. Additionally, all learned distributions are also regularized using a KL-divergence with the standard normal distribution $\mathcal{N}(\mathbf{0}, \mathbf{1})$. This is optimized using constraint optimization, where the Lagrange multiplier weights $\lambda_i$ are scaled automatically for the terms to stay below predefined constraints (Rezende & Viola, 2018).

$$
\begin{aligned}
L = \sum_{t=0}^{T} & \lambda_{x,t}||\mathbf{x}_t - p_\phi(\mathbf{x}_t|\mathbf{p}_t, i)||_2 + \lambda_{i,t} D_{KL}[q_\phi(i|\mathbf{x}_{t,m})||p(i)] + \lambda_{m,t}||m_\phi(\mathbf{x}_t) - \mathbf{m}_t||_2 + \\
& D_{KL}[q_\phi(\mathbf{p}_t|\mathbf{x}_{t,m})||\mathcal{N}(\mathbf{0}, \mathbf{1})] + \\
\sum_{t=1}^{T} & \lambda_{\text{trans},t}||\mathbf{x}_t - p_\phi(\mathbf{x}_t|\mathbf{p}_{t,\text{trans}})||_2 + D_{KL}[q_\phi(\mathbf{p}_t|\mathbf{x}_{t,m})||p_\phi(\mathbf{p}_{t,\text{trans}}|\mathbf{p}_{t-1}, \mathbf{a}_{t-1})] + \\
& D_{KL}[p_\phi(\mathbf{p}_{t,\text{trans}}|\mathbf{p}_{t-1})||\mathcal{N}(\mathbf{0}, \mathbf{1})]
\end{aligned}
\tag{1}
$$

## A.2 Additional Qualitative Results

In Figure 4, we show a prediction of a sequence where the agent has seen multiple observations of the object without occlusion. We can first observe that in the CCN-based model, correct inference has been made about the presence of the sugar box, even though it has only been partially observed, while the other models do not have enough information to start reconstructing the sugar box. It can also be observed that when the agent has a large set of evidence about the occluded object and/or the object is not completely occluded, it will have a better prediction and does not forget the object, also in case of the SSM and RSSM models.

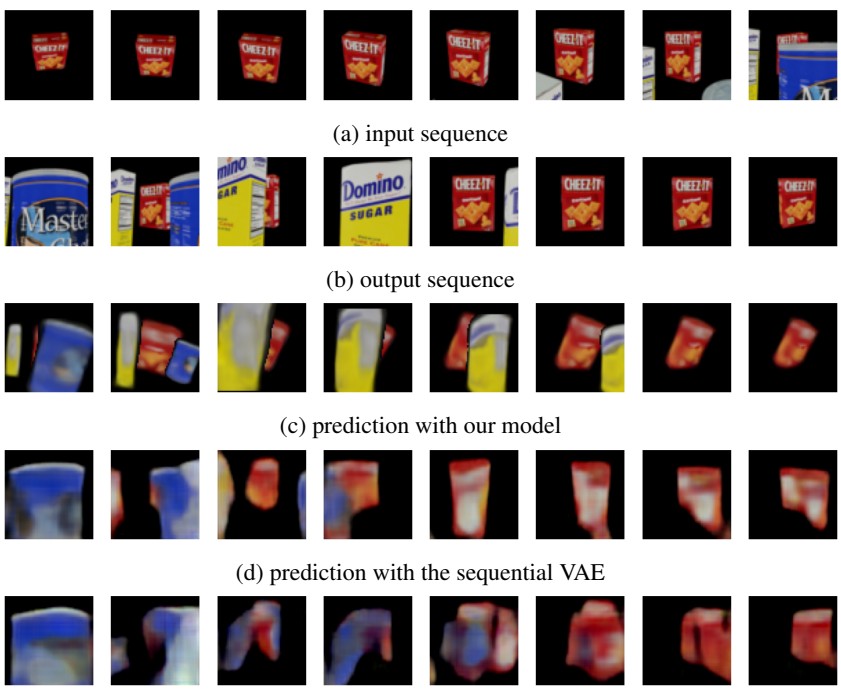

Figure 4: Sequence of observations where an object is only partially occluded in the final frame, together with predictions by the different models.

