# OpenReview forum: "Enforcing Object Permanence using Hierarchical Object-Centric Generative Models"
_NeurIPS.cc/2022/Workshop/SVRHM — SVRHM Poster_

### Official Review · Reviewer_x6YN · 2022-10-08
**Review: Object permenance using object-centric generative models**

**Rating:** 6
**Confidence:** 4

**Review:**

**Summary**: The authors propose an object-centric generative model which appears to show better results at reconstructing occluded objects conditioned on actions, indicating some notion of object-permenance.

**Positives**: The method appears to show positive results with respect to the authors stated goal and an interesting discussion of related work is provided.

**Limitations**: The description of the model architecture is insufficient in the main text especially considering this is the main focus of the proposed method. It is further unclear why the authors chose to put the ELBO in the appendix when this could have easily fit in the main text and still been below the page limit. Finally, although the presented reconstructions are promising, it is not clear if these are cherry-picked given only a single example is presented. The adjusted MSE loss is a welcomed addition, given this presumably summarizes performance over the whole dataset, however since this loss is also non-standard, it is a bit challenging to evaluate objectively. Overall, I recommend the authors include additional model and experimental details in their paper to help readers better understand their contribution.

**Minor suggestions**: The ELBO in equation (1) appears to slightly confuse the function which is used to parameterize the mean of the generative distribution, with the distribution itself (at least in notation). This is different from how such losses are typically presented in the generative modeling community, and I thus recommend the authors review the related literature to understand this difference.

**Conclusion**: I find the paper to be borderline. There are many limitations with the presentation of the work itself and the evaluation, but I find no inherent inaccuracies in the statements made, and thus find it to be acceptable as preliminary work for a workshop. For a more refined submission I recommend the authors elaborate on all methods used and increase the perceived reliability of their results with further validation.

---

### Official Review · Reviewer_Vjq6 · 2022-10-12
**Novel approach, environment, and metrics for testing object permanence in generative vision models**

**Rating:** 7
**Confidence:** 3

**Review:**

**Summary:**

The authors propose a framework to test object permanence in generative vision models. They create a 3D simulation environment based on the YCB dataset, where a sequential model sees first sixteen frames with an object going into occlusion, with the aim being to generate the occluded object in the future sixteen frames as the camera pans back to it.

**Strengths:**

1. The problem is very well motivated, and the proposed measure of object permanence (distance scaled MSE for the pixels corresponding to occluded object, as well as detection rate) is well formulated and thought out.

2. The idea of having a separate object-centric representation and a global camera-view scene representation seems quite relevant to the problem of learning representations with object permanence, and proves effective in the author's evaluations.

**Weaknesses:**

The method is hard to generalize outside of simulated environments where pixel/voxel level object masks are not available. This greatly reduces the utlity of this approach for simple tasks where only semantic labels at scene or object level are available.

**Disclaimer:** The reviewer is unfamiliar with the prior works on object-centric generative models which the authors built upon, and hence was not able to follow certain descriptions of how the model is constructed and trained. It would be worthwhile to add an appendix describing more details about the model and training procedure so that the paper is more self-contained.

---

### Official Review · Reviewer_RWuS · 2022-10-14

**Rating:** 6
**Confidence:** 4

**Review:**

This work investigates whether object-centric models show signs of object permanence and tried to improve this and make them behave more similar to humans.

## Pros
- The authors generate a new dataset to test their model on.
- This work introduces a new model, investigates and compares it to other models.

## Cons/Questions
- The related work seems very limited and misses a lot of relevant work on object-centric models.
- The reasoning why the two baselines (SSM and RSSM) were chosen and other object-centric models have been ignored (e.g, [1-3]).
- The definition of the model in Section 2 is a bit unclear: Since there is only a single mask $m_\phi$ it is unclear to me how this approach would be applied to scenes with multiple objects (which is crucial if one wants to investigate the object-permanence properties of the network). I assume for the multi-object case you instead have masks $m_{i, \phi}$ and then sum over different object indices i? Also, the variable $a_t$ is never defined - is it correct that this is the action? Since the model is the centerpience of this work, I encourage the authors to make the description of the model as clear and crisp as possible.
- Figure 2: The caption makes it sounds as if this was the only way object-centric models can work, although this just depicts a special instantiation of this class of models. Also, the caption is confusing as you say "Using the location [...] the location can be estimated" which seems trivial.
- L75: Do the objects all have the same size, i.e., is it only three different objects, or does the dataset contain scaled versions of the objects, too?
- L93: How was this threshold chosen? If the aim is to investigate how well the model deals with occluded objects it might make sense to disentangle this property from its general ability to reconstruct/generate images. For example, one could normalize the prediction error in the last frame with the average prediction error before calculating the "detected" metric. This way, you are not measuing how well a model reconstructs the last image, but instead how well it reconstructs the last image compared to how well it can reconstruct images in genera, which seems to be closer related to testing for object permanence.


[1] Singh, Gautam, et al. "Simple Unsupervised Object-Centric Learning for Complex and Naturalistic Videos". arXiv prepring 2205.14065 (2022) \
[2] Kipf, Thomas, et al. "Conditional Object-Centric Learning from Video." International Conference on Learning Representations. 2021. \
[3]  Elsayed, Gamaleldin F., et al. "Savi++: Towards end-to-end object-centric learning from real-world videos." arXiv preprint arXiv:2206.07764 (2022).

---

### Official Review · Reviewer_it9i · 2022-10-16
**Potentially interesting model but significance unclear**

**Rating:** 5
**Confidence:** 4

**Review:**

The authors propose an object-centric generative model as a model of object permanence. An object-specific generative component is first pre-trained on different views of a single object. The object-centric generative model then incorporates this pre-trained component and is trained to predict the next frame of a video of multiple objects - one of which becomes temporarily occluded due to camera motion. The authors then show that the model predicts video frames after an occlusion event had occurred and the occluded object reappears from occlusion.

The paper makes a contribution to a larger field of similar models. The manuscript is mostly written clearly, but clarity is lacking when it comes to critical aspects of the paper, namely the model description and the description of the evaluation methods. Given the existing literature on object-centric generative models (which is not cited), this paper's originality and significance do not stand out or are at least not conveyed clearly enough. After reading the paper I felt a bit disappointed mainly because I feel it left me with very few new insights even though the proposed model appears interesting.

1) How does this relate to the many existing object-centric generative models? There is little embedding into the existing literature. For example, the relevant literature on unsupervised scene representation learning is not cited and its relation to the current work is not discussed (see for example last year's oral presentation by Arora et al.). There are a lot of very similar object-centric generative models already from several years ago. How is this one different and addresses a gap in the existing literature?

2) What can we learn conceptually from the model? The authors compare their model to only two other models in a quantitative way (a state space model by Çatal et al., 2020, and a recurrent state space model by Hafner et al., 2020). The reader is left a bit clueless about what to glean from that comparison. How do SSM and RSSM differ conceptually from the proposed model and which conceptual differences are relevant for the performance difference? Lesion studies or variants of the proposed model with changed components (e.g., end-to-end training rather than object-centric pre-training) would have been another way to provide conceptual insights.

3) There are not enough details provided to understand the model. In particular, explaining the symbols and tying them in with the text would go a long way (there was still enough space). Some examples: Is $i$ an index (as the non-bold style suggests) or the vector encoding the object-centric representation? How is the global scene representation structured? The text suggests that the global scene representation is a concatenation of object identities and positions but the vector $\mathbf{p}_t$ (rather than a set of $\mathbf{p}_m,_t$) seems to imply something more complicated than this.

4) What encourages the model to retain the object representation of an object during the occlusion? The model is trained to predict transitions between pairs of frames. There is nothing that encourages the model to retain an object representation through longer stretches of occlusion. At the very minimum, one would need three frames, (1) object present, (2) object occluded, and (3) object reappears. In contrast, the regularization of the object representation towards the prior (second D_{KL}-term in eq (1) in the appendix) might push the representation of an occluded object towards the prior, i.e., lead to "forgetting" of the invisible object. It would be good to see whether the model "breaks" if the occlusion duration becomes longer.